# The War Is on: The Immune System against Glioblastoma—How Can NK Cells Drive This Battle?

**DOI:** 10.3390/biomedicines10020400

**Published:** 2022-02-08

**Authors:** Lucas Henrique Rodrigues da Silva, Luana Correia Croda Catharino, Viviane Jennifer da Silva, Gabriela Coeli Menezes Evangelista, José Alexandre Marzagão Barbuto

**Affiliations:** 1Departamento de Imunologia, Instituto de Ciencias Biomedicas, Universidade de Sao Paulo, Sao Paulo 05508000, Brazil; lucashenri17@usp.br (L.H.R.d.S.); luana_correia@usp.br (L.C.C.C.); viviane17jennifer@gmail.com (V.J.d.S.); gabrielacoeli@usp.br (G.C.M.E.); 2Laboratory of Medical Investigation in Pathogenesis and Targeted Therapy in Onco-Immuno-Hematology (LIM-31), Departamento de Hematologia, Hospital das Clínicas HCFMUSP, Faculdade de Medicina, Universidade de Sao Paulo, Sao Paulo 0124690, Brazil

**Keywords:** immunotherapy, natural killer cells, cancer, glioblastoma

## Abstract

Natural killer (NK) cells are innate lymphocytes that play an important role in immunosurveillance, acting alongside other immune cells in the response against various types of malignant tumors and the prevention of metastasis. Since their discovery in the 1970s, they have been thoroughly studied for their capacity to kill neoplastic cells without the need for previous sensitization, executing rapid and robust cytotoxic activity, but also helper functions. In agreement with this, NK cells are being exploited in many ways to treat cancer. The broad arsenal of NK-based therapies includes adoptive transfer of in vitro expanded and activated cells, genetically engineered cells to contain chimeric antigen receptors (CAR-NKs), in vivo stimulation of NK cells (by cytokine therapy, checkpoint blockade therapies, etc.), and tumor-specific antibody-guided NK cells, among others. In this article, we review pivotal aspects of NK cells’ biology and their contribution to immune responses against tumors, as well as providing a wide perspective on the many antineoplastic strategies using NK cells. Finally, we also discuss those approaches that have the potential to control glioblastoma—a disease that, currently, causes inevitable death, usually in a short time after diagnosis.

## 1. Introduction

Cancers are a heterogeneous and complex set of diseases that pose a great concern to public health, comprising one of the predominant causes of death worldwide. An estimate from the Global Cancer Observatory (GLOBOCAN) indicates that in 2020 alone these diseases were responsible for the deaths of approximately 10 million people [1]. A projection for 2040 suggests that around 30.2 million people will be affected, and 16.3 million deaths will occur [1].

Glioblastoma (GBM) is the primary cancer with the highest incidence within the central nervous system, and its presence considerably reduces patients’ quality of life [2]. Its current treatment—including maximal surgical resection, adjuvant radiotherapy, and chemotherapy with temozolomide—significantly improves patient survival [3]. However, this therapeutic regimen is still not enough, as half of the patients die within approximately 15 months after diagnosis [4], and less than 5% of them remain alive after 5 years [5]. Relapse occurs frequently, and death is inevitable.

A broad arsenal of treatments has been established to control or potentially cure neoplasms; more recently, immunotherapy has emerged as a new therapeutic pillar in oncology. Within this treatment modality, most strategies have been centered on either antibodies or T cells, but lately, new studies have sought to take advantage of the antitumor activity of NK cells.

In this review, we first aim to provide aspects of basilar NK cell biology, focusing on their contribution to antitumor surveillance and response. Then, we consider what is known about their functional status in GBM, highlighting the mechanisms of immunosuppression acting in this disease. Finally, we present in more detail the state of the art of immunotherapeutic NK-based strategies against GBM, comprising both already-used strategies and strategies that have shown promising results for the treatment of GBM in preclinical trials.

## 2. Natural Killer Cells

Discovered in the 1970s during cytotoxicity studies of lymphocytes, natural killer (NK) cells were initially characterized as large granular lymphocytes with “natural” cytotoxicity, since they did not require prior exposure to their targets in order to perform their cytotoxic activity [6,7,8]. They were also defined as “null” cells, considering that they were non-adherent cells in which the genes encoding antigen receptors were not reorganized, resulting in the Ig^−^ phenotype and in their inability to perform rosette formation with erythrocytes (non-B or -T cells) [9]. Finally, NK cells were understood as cells with cytotoxic capacity against a wide range of target cells, but also as cells that play significant roles in inflammation and in innate and acquired immune responses, via their production of chemokines and cytokines [10,11,12].

Kiessling and Kärre’s studies paved the way for the formulation of the “missing self” hypothesis for the target recognition of NK cells. These works demonstrated that murine lymphoma cells with low or absent expression of genes encoded by the major histocompatibility complex (MHC) class I were lysed very efficiently by NK cells, while cells expressing such molecules were resistant to lysis [13,14]. A conclusion of these experiments and a necessary factor for the “missing self” model is that NK cells should express at least one specific receptor for MHC-encoded class I molecules (MHC I) in order to be able to detect the “self” and its absence. The discovery of MHC-specific inhibitory receptors provided the first explanation for the “tolerance” of NK cells to the “normal” cells within the body (reviewed by [15]).

Today, NK cells are classified as belonging to the group 1 of innate lymphoid cells (ILC1), as they do not depend on clonal receptors to recognize their targets, and can produce IFN-γ [16]. In addition to being present in the peripheral blood, representing ~5–15% of the lymphocytes, they are also distributed in several lymphoid and non-lymphoid tissues, such as the lymph nodes, tonsils, spleen, liver, intestines, bone marrow, and gravid uterus [17,18]. Despite NK cells being classified as innate cells, NK cell responses can exhibit the adaptive phenotype of immunological memory or trained immunity under circumstances such as viral infections, hapten challenge, and stimulation with IL-12, IL-15, and IL-18 cytokines. Interestingly, memory or “adaptive” NK cells present a long lifespan and perform improved responses upon re-stimulation [19]. 

The phenotypic identification of NK cells within the leukocyte population is distinct in murine and human models. In mice, they can be identified by the absence of CD3 and the presence of NK1.1—an epitope shared by the murine NKPR1b and NKPR1c receptors [20]; however, considering the existence of murine NK1.1^−^ strains [21], other phenotypes that are also used include the CD3^−^ CD49b^+^ or CD3^−^ NKp46^+^ [20]. In humans, NK cells are mainly identified as CD3^−^ CD56^+^ cells [22,23].

NK cell development occurs in the bone marrow, and is thought to be a linear process (controversies remain at this point [24]) with multiple stages, characterized by different membrane markers (Figure 1); it starts from a hematopoietic stem cell (HSC) that self-renews and differentiates into a compromised multipotent lymphoid progenitor (MLP). This gives rise to the common lymphoid progenitor (CLP)—a cell with the potential to differentiate between the lineages of B, T, and NK lymphocytes [25,26,27,28,29,30]. Upon the expression of CD122 (IL-2Rβ IL-15Rβ), the lymphoid progenitor becomes responsive to IL-15 [27,31,32]. IL-15 is recognized as an important cytokine for the development, maturation, homeostasis, and survival of NK cells, since mice deficient in IL-15 have lower amounts of mature NK cells [33,34,35,36]. This cytokine promotes the generation of NK cell precursors (NKPs) and immature NK cells (iNKs). The appearance of CD56 (NCAM) indicates the final transition from iNKs to mature NKs (mNKs) [37,38]. Bone marrow exit occurs after the functional gain of cytotoxicity via the expression of NKp46 (CD335) and NKG2D receptors, and at the end of maturation in secondary lymphoid organs, NK cells become positive for CD43 and Mac-1 [39]. 

Although they do not have V(D)J somatic recombination-generated antigen receptors, unlike B and T lymphocytes, NK cells are endowed with a myriad of germline-encoded receptors that act as immunological sensors, recognizing a wide range of molecular signals [40]. In addition to recognition, these receptors play a crucial role in controlling cell activation, and are therefore categorized as having activating and inhibitory functions [41] (Table 1).

Activating receptors include natural cytotoxicity receptors (NCRs), such as NKp30, NKp44, and NKp46. Although most of the endogenous ligands of these receptors are unknown, it has been noted in recent years that NCR interactions are extremely heterogeneous, and may involve viral ligands, proteins expressed upon stress, surface glycoproteins, altered matrix molecules, surface-exposed nuclear molecules, and enzymatically released ligands [42,43]. Other activating receptors include the FcγRIII receptor (CD16), which recognizes IgG antibodies [44], and toll-like receptors, which recognize molecular patterns associated with damage and pathogens (DAMPs and PAMPs, respectively) [57]. 

Inhibitory receptors include molecules such as PD-1, CTLA-4, TIGIT, LAG-3, and TIM3, described as immune checkpoints that impair NK cell activation [51]. There are also families with mixed-function receptors, which can promote and suppress NK activation, such as the killer cell immunoglobulin-like receptors (KIRs) [40,49,58] and type-C lectin-like NKG2 family [47,48]. KIRs generally recognize class I major histocompatibility complex molecules—mainly HLA-A, B, and C classical MHC molecules—and can be activating or inhibitory [49]. In turn, NKG2 receptors mainly recognize molecules of increased expression under stress, such as MICA, MICB, and ULBP (recognized by NKG2D), in addition to the non-conventional MHC molecule HLA-E (recognized by CD94/NKG2A, C) [47,48,58]. The DNAX accessory molecule-1 (DNAM-1), Tactile (CD96), and TIGIT are members of the immunoglobulin superfamily which, as demonstrated in Table 1, have one common ligand—the poliovirus receptor (PVR, CD155), which displays binding with different affinities, with the strongest interaction with TIGIT, followed by CD96, and the weakest with DNAM-1. PVR is overexpressed in several tumors; thus, the expression level of each receptor in NK cells is thought to determine the functional role of PVR in NK activation [45,56]. DNAM-1 is well established as a co-activating receptor, and has important functions in tumor immunosurveillance, given that its absence in knockout mice favors the incidence and growth of carcinogen-induced tumors, as well as reducing the clearance of CD155^+^ transplanted tumor cells in vivo and the cytotoxic activity of NK and T cells in vitro [59]. The functional role of CD96 is controversial; despite studies showing correlations between its expression and the activation of NK cells [60] and CD8+ T cells [61], beneficial roles of its neutralization or knockout have also been described, such as metastasis suppression via IFN-γ secretion by murine NK cells [62,63]; in fact, its co-inhibitory function is being explored in antibody-dependent immune checkpoint blockade strategies [51]. In addition to the already-mentioned receptors, NK cells contain SLAM family receptors [50], chemokine receptors [64], and others [43,58].

Ligand specificity and functional differences in NK receptors suggest an astonishing heterogeneity of activating and inhibitory stimuli underway in NK cells present in biological tissues. Consequently, it is not surprising that the dynamic integration of these signals determines the fate of NK cell activation and responsiveness [40,65]. Over the years, a consensus has been established on the fact that activation stimuli may not always culminate in an effector response, as several factors modulate the responsiveness of NK cells, from their development to even after their functional maturation (reviewed by [65]).

The pattern expression of the aforementioned receptors generates a great diversity regarding types of NK cells. The KIR receptor genes, for example, in addition to being highly polymorphic, are stochastically expressed in the NK cell population [66]. Due to these factors, it is estimated that ~20% of CD56^dim^ NK cells in peripheral blood do not have at least one inhibitory KIR or NKG2A receptor capable of recognizing some MHC I molecules [67]. Therefore, in order to avoid self-reactivity, NK cells undergo a process called “licensing” or “education” during their development, where only NK cells capable of recognizing MHC I alleles become functionally mature [66]. In agreement with this, studies over the past decade involving NK cell populations in humans or mice have revealed that a considerable portion of cells do not express specific inhibitory MHC I receptors, but are still self-tolerant [68,69,70].

The cytotoxic effect involves the formation of an immunological synapse with the target cell, followed by exocytosis of cytoplasmic lytic granules containing perforin 1 and granzymes, which together result in the formation of pores in the target cell membrane and initiation of the programmed cell death pathway [71]. Another consequence of degranulation is the increase in the expression of the Fas ligand protein (FasL, CD95L) in the cell surface [72,73]. FasL is recognized by the transmembrane protein Fas (CD95)—a TNF family death receptor that activates the extrinsic pathway of apoptosis. This signaling mechanism is shared by other receptors of the TNF family, such as the functional receptors TRAIL R1 and TRAIL R2, which recognize the TRAIL protein expressed in activated NK cells [71].

In addition to cytotoxicity, it has been increasingly recognized that NK cells establish important immunomodulatory communications with other innate and adaptive cells. This crosstalk occurs through physical contact [74], or via secretion of soluble factors such as cytokines, and may eventually, be recognized as the main physiological function of these cells.

There seems to be a hierarchy of stimulatory signal strength to trigger each of these actions: the formation of an LFA-1-mediated immunological synapse requires the weakest activation signal; the secretion of chemokines, such as MIP-1, involves a stronger signal; degranulation and cytotoxicity require an even stronger signal; and the production of cytokines, such as IFN-γ and TNF-α, necessitates the most stringent signal [40]. Through these mechanisms, NK cells play pivotal roles in numerous physiological and pathological conditions. Their participation during embryonic pregnancy [75], as well as in autoinflammatory diseases [41,76], immunodeficiency [77], neurodegenerative diseases [64,78], autoimmune diseases [79], possibly allergic diseases [80], and during the control of intracellular infections, such as viral infections [81,82], among several other situations, is remarkable. However, since their discovery in the 1970s [83], they have been especially appreciated for their ability to recognize and kill neoplastic cells.

## 3. NK Cells in Immunosurveillance against Cancer

At least in murine models, NK cells play a central role in immunosurveillance, acting against established tumors and in the prevention of metastasis, conferring them an attractive role in immune strategies aiming to control the progression of tumors and potentially eradicating them [84,85]. Their participation in tumor immunosurveillance has been evidenced in several studies, and is well established in murine models, where blocking the activity of effector mechanisms shared by NK cells—such as perforin, TRAIL, and IFN-γ—leads to a higher incidence and/or development of tumors when compared to wild animals [86,87,88,89,90,91]. Likewise, and pointing more specifically to NK cells, their depletion by treatment with monoclonal antibodies to asialo-GM1 or NK1.1 increases the susceptibility of mice to chemically induced fibrosarcomas [92].

Despite providing strong evidence for a significant role of NK cells in immunosurveillance, these are studies in murine models, and their equivalence in humans could be questioned. In this sense, a series of observations can be brought to argue in favor of an equivalent role of NK cells in humans. Initially, it must be recognized that NK cells from healthy individuals do exert ex vivo antitumor activity against cells of numerous primary tumor lineages, demonstrating the existence of spontaneous cytotoxicity against human tumors [93]. Another pivotal piece of evidence is that the functional activity of NK cells significantly impacts the incidence of cancers. This was demonstrated by a prospective observational study, with an 11-year follow-up, carried out from 1986 onwards in a Japanese population [94]. The study looked at the natural cytotoxicity of peripheral blood cells from 3.625 healthy individuals against the K-562 tumor lineage (attributed primarily to NK cells) and, after adjustment for age, sex, and lifestyle, showed that lower levels of cytotoxicity were correlated with a higher incidence of cancer, while medium or high levels of cytotoxicity were protective factors, decreasing the relative risk of tumor incidence [94].

The roles of the immune system’s different components in immunosurveillance can be inferred by observing patients with specific immunodeficiencies. However, for the establishment of the role of NK cells, the rarity of well-defined and specific immunodeficiencies of NK cells is a major obstacle [95]. Ebbo et al. (2016) [96], analyzing 457 patients with common variable immunodeficiency (CVID) in a 9-year follow-up multicenter cohort study, showed that the degree of NK lymphopenia did not correlate with the incidence of cancer; they hypothesized that the lack of correlation could be attributed to patients’ median age in the evaluation period and, moreover, that only a small fraction of the patients had isolated NK cell immunodeficiencies. Classical NK cell deficiency (CNKD), on the other hand, is a severe condition of immunodeficiency with NK-cell-only lymphopenia, where NK cells are low or absent in the peripheral blood. Among the few reported cases of CNKD, the development of cancers (mainly leukemias and human papilloma virus (HPV)- and Epstein–Barr virus (EBV)-associated cancers) has been reported [77]. Despite this observation, the difficulty faced by these patients in controlling or resolving infections, which generate carcinogenesis-contributing factors [94], precludes the attribution of a definite role for NK cells in immunosurveillance. Another confounding factor in these patients is that that *GATA2* and *MCM4*—the two main genes involved in the origin of CNKD—could be relevant in carcinogenesis [97,98], including *GATA2* mutations identified in families with predisposition to myelodysplastic syndrome and acute myeloid leukemia (MDS-AML) [97]. 

Nevertheless, when considering a possible role for NK cells in immunosurveillance, several factors must be added to the equation, since they can influence the outcome significantly; these include the distribution of cell subsets in different types of tissues, the abundance of NK cells in the tumor, the genetic constitution of the components of each patient’s immune system, and the evasion strategies employed by the tumor and the microenvironment to modulate the efficacy of NK cells (and other immune cells) [99]. Though this complexity may make it difficult to ascertain the contribution of NK cells to immunosurveillance, it points to different lines of investigation that might clarify the role these cells and indicate pathways to effectively exploit their anticancer potential.

### 3.1. Tumor Cell Recognition

NK cells can kill cancer cells while maintaining tolerance against healthy cells [14]. As already mentioned, any selective recognition by NK cells depends on the signals integrated by their activating and inhibitory receptors. Unlike most healthy cells, tumor cells may have low or reduced expression levels of the MHC-I molecules on their surface; this prevents the engagement of inhibitory receptors—such as KIR and NKG2A receptors—which would enable NK cell activation over the target cell [100,101,102]. In parallel, tumor cells can express induced ligands—such as MICA, MICB, B7-H6, CD58, and ICAM1—which engage activating receptors [101]. Indirect recognition through the CD16 receptor, which binds to the Fc portion of IgG antibodies associated with tumor antigens, is also possible, stimulating antibody-mediated cytotoxicity [103]. The expression level of these molecules in tumor and NK cells dictates the malignant cell’s death processes [101]. However, NK cell activity can be altered by numerous components of the tumor microenvironment, including immunosuppressive cytokines, such as TGF-β, and pro-inflammatory cytokines, such as IL-12, IL-15, IL-18, and IL-21 [101]. Hypoxia is one major factor also implicated in immune regulation, through the expression of the hypoxia-inducible factor 1α (HIF-1α)—a transcriptional factor that promotes multiple signaling, inducing immune suppression, including that of NK cells [104,105].

### 3.2. Antitumor Functions

Upon activation, NK cells can exert antitumor activities in two ways: by directly killing malignant cells, or by modulating the activity of other leukocytes (Figure 2). The direct killing is thought to be mediated mainly by perforin/granzyme-dependent cytotoxicity, as noted in numerous experimental models [85,106]. Moreover, an important role of cytotoxicity mediated by the death receptors FasL and TRAIL has also been observed [107]. Time-lapse microscopy analyses indicate that a single NK cell can kill multiple tumor cells in a serial manner [106,108]. In the serial killing of neoplastic cells susceptible to both death mechanisms, the granzyme-mediated death initially predominates, and the contribution of the pathway dependent on death receptors progressively increases [106]. 

In addition to cytotoxicity, NK cells can produce and secrete soluble factors, being a major source of IFN-γ, CCL5, XCL1, XCL2 and Flt3L, and GM-CSF; through these cytokines and chemokines, they establish a cooperative relationship with other immune cells, and can indirectly trigger the death of tumor cells [109]. One important chemokine axis regulating NK function, for example, is the “regulated on activation, normal T-cells expressed and secreted” (RANTES), also known as CCL5 [110,111]. This chemokine, which is secreted by immune cells such as NK and T cells, has the ability to regulate the infiltration of multiple immune cells in inflammatory focus, including NK cells, T cells and monocytes [112,113], in addition to contributing to NK cell expansion and activation [114]. On the other hand, through the cytokine IFN-γ, NK cells can activate macrophages, directing them to kill tumor cells [109], in addition to inducing the differentiation of Th1 lymphocytes in lymph nodes [115]. In a murine melanoma model, NK cells secreted the chemokines CCL5 and XCL1, recruiting cDC1 dendritic cells into the tumor microenvironment, leading to controlled tumor growth [116]. A similar dynamic of control of the cDC1 levels and improved prognosis was linked to Flt3L cytokine production by intratumoral murine NK cells [117]. This DC subtype plays a key role in inducing the cytotoxic CD8^+^ T cell (LTCD8) response against tumor antigens in murine models [118,119]. Significantly, the NK cell genetic signature profile also correlates with the cDC1 cell profile in several human cancers, and both are associated with the levels of transcripts of the chemokines CCL5, XCL1, and XCL2, suggesting an equivalent role for NK cells in humans [116]. 

Another observed activity of NK cells is the induction of the maturation of dendritic cells, both through physical contact via NKp30, and through the secretion of cytokines such as IFN-γ and GM-CSF [16]. In vitro, IL-2-activated NK cells can facilitate the presentation of tumor antigens by dendritic cells via MHC-I, improving the activation of Ag-specific CD8^+^ T lymphocytes [120]. Indeed, NK-DC crosstalk results in the in vivo formation of a protective LTCD8 response against the tumor [121]. In agreement with the multiple effects of NK cells in modulating adaptive antitumor responses, in lymphoma-bearing mice, the presence of NK cells or the cytokine IFN-γ is essential for the generation of a long-lasting adaptive (Th1 and CTL) antitumor response [122].

Remarkably, NK cell infiltration and activity can predict the trajectory of different types of human cancers [101]. Nersesian et al. (2021) [123] performed a systematic review of 53 scientific articles, comprising a total of 9624 patients carrying different types of solid tumors, including head and neck, liver, lung, endometrial, breast, and 10 other types of tumors. Most of the tumors assessed (~59.3%) showed a positive association between NK cell infiltration and longer overall survival, while ~38.9% showed no correlation and ~1.9% pointed to a worse prognosis. A meta-analysis of the data showed that NK cell infiltration is associated with a lower risk of death. This suggests an important contribution of these cells to the natural resistance against solid tumors, and indicates their potential for immunotherapy.

## 4. NK Cells in GBM

The dynamic blood–brain barrier (BBB) and other factors maintain a reversible state of immune privilege in the brain parenchyma, and prevent the entry of most peripheral blood immune cells into the non-inflamed central nervous system [124]. However, glioblastoma cells disrupt astrocyte extensions involving blood vessels and modify vascular homeostasis, thus breaking the BBB [125]. In addition, they maintain a pro-angiogenic and pro-inflammatory environment that leads to increased tissue permeability and chemotaxis. These changes support the recruitment of diverse immune cells into the tumor, which can be identified by immunohistochemical or flow cytometry techniques [126,127,128,129,130]. It is also possible to obtain the sequences of the total RNA in GBM tissue stored in databases, and later use bioinformatics tools, such as CIBERSORT, to deconvolute immune populations according to the profiles of genetic signatures [131,132]. Collectively, these strategies evidence the presence of diverse leukocyte populations, including macrophages, CD4^+^ T lymphocytes, CD8^+^ T lymphocytes, B lymphocytes, neutrophils, dendritic cells, mast cells, and other immune cells, including the NK cells [126,127,128,129,130,132].

Few attempts to assess NK cells in the GBM microenvironment have been described in the literature. Furthermore, a comparative analysis between these studies is difficult, since different methods of analysis, cell markers, and gating strategies of the NK cell population have been employed. It is noteworthy that in some cases the number of samples is low [128,132], and observations suggest the existence of variations between individuals [127,129,132]. With all of this in mind, some reports indicate that in GBM, a low frequency of phenotypically and functionally altered infiltrating NK cells can be found (Table 2).

One may suggest that alterations such as reduced transcription of IFN-γ [132], decreased expression of some activating receptors (e.g., NKp30, NKG2D, DNAM-1, and CD2) [133], and the expression of transcripts associated with quiescent NK cells in GBM tissue [132] collectively represent tumor-modified NK cells, but the functional impact of these changes remains elusive in GBM. The NK CD56^dim^ CD16^−^ phenotype, shown to be the predominant NK type in a report [128], could be generated by the loss of membrane expression of CD16, which is a process mediated by the membrane ADAM-17 metalloproteinase, which occurs in vitro under various activation stimuli (e.g., cytokines, activating antibodies, and target cells) [135,136]. Curiously, in a 2019 study, CD56^dim^ CD16^−^ NK cells were reported as the predominant NK subtype (86%) infiltrating the melanoma tumor microenvironment, and their peripheral blood count was positively correlated with decreased cytotoxicity against K-562 tumor targets [137]. Another phenotypic change described in NK cells in GBM patients was the lower expression of the chemokine receptor CXCR3 in peripheral blood NK cells compared to tumor NK cells [132]; this could impair the infiltration in the tumor microenvironment, considering that CXCR3 expression in NK cells has been shown to be crucial for this process in other tumor models in mice [138]. However, additional studies are required in order to confirm these findings and their potential role in the anti-GBM response exerted by NK.

In a BALB/c-nude mouse orthotopic GBM model, depletion of NK cells by the administration of anti-NK1.1 antibodies did not affect the progression of primary tumor mass, but was associated with the appearance of lung metastases, which were absent when these T-cell-deficient mice maintained their NK cells in peripheral blood [139]. This result is consistent with the prevalent view of NK cells’ role in metastasis control and, although speculatively, could be interpreted as indicating that NK cells, in addition to many other potential contributing factors [140], might be involved in the observed absence of extracranial GBM metastases in humans. 

Some studies, noting low intratumoral infiltration in brain cancers, hypothesized that NK cells would not play a significant role in the natural control of these tumors [141]. In agreement with this, the absence of correlation between the accumulation of NK cells in the tumor and the prognosis of patients with GBM [129,142] has been reported. In contrast, independent correlation between activated NK cells and worse [143] or improved [144] prognosis has been also described. These conflicting data are probably related to the great heterogeneity of tumors classified as GBM [129], and to the various escape mechanisms that might be present in these tumors [145], pointing to the need for a better and more complete characterization of immune infiltration in GBM before one can imply a specific role for NK cells—or any other cells, for that matter—in its biology.

In fact, many known mechanisms could counteract NK antitumor activities within the GBM. These include intrinsic characteristics of neoplastic cells and factors dependent on the tumor microenvironment, such as structural elements and the composition of the immune infiltrate [146]. For example, GBM cells can maintain the expression of class I MHC molecules such as HLA-A, B, and C [128], which promote the engagement of inhibitory KIR (iKIR) receptors and prevent NK cell cytotoxicity. GBM cells may fail to express MIC-A/B and CD70- ligands to activate receptors NKG2D and CD27 on NK cells, respectively [128]. The lectin-like transcript 1—a CD161 receptor ligand—has been described in glioma lines, and suppresses NK cell cytotoxicity [147]. The HLA-E protein is frequently overexpressed GBM in vivo and in vitro, and its presence in GBM tumor cells has been shown to impair NK cytotoxicity in an NKG2A/CD94-dependent manner [148]. In addition, GBM cells can secrete immunosuppressive factors such as TGF-β [149], which may act locally and systemically, impairing the activation, cytotoxicity, proliferation, and production of IFN-γ by NK cells [150]. In addition, TGF-β reduces the expression of activating receptors on NK cells, such as NKp30, DNAM-1, and NKG2D [151,152]. Although the role of NK cells in GBM control is still uncertain, immunotherapeutic strategies attempting to overcome known NK-cell-suppression mechanisms in this disease may effectively harness these cells to fight it.

## 5. NK-Cell-Based Immunotherapy

Cancer immunotherapy has been an elusive goal ever since the proposition of the immunosurveillance hypothesis by Burnet [153]. However, due to continuous elucidation of immune and tumor-evasion mechanisms, in recent years there has been a significant change in this previously dismal picture, and cancer immunotherapy has become a concrete option for many patients with various types of cancer [154].

Due to the many roles that NK cells can play against cancer, as well as their ability to quickly recognize and attack cancer cells, without the need for previous sensitization [155], while also presenting limited reactivity against healthy tissues [156], many studies have explored the potential of NK cells in cancer immunotherapy. NK-cell-based immunotherapy presents the possibility of targeting tumors that still lack well-defined antigens for specific response, as well as the possibility of using allogeneic products prepared in advance, and that may be administered in multiple patients without causing graft-versus-host disease (GvHD) [157,158,159], thus potentially leading to less toxicity in comparison to CAR T-cell infusions, for example [156,159].

Different strategies of NK-cell-based immunotherapy for GBM are summarized in Figure 3; they pursue different strategies, aiming at the number or in vivo persistence of these cells [160], the activation or blocking of their inhibitory signals, their cytotoxic potential through adoptive cell therapies [155], or sensitizing cancer cells to NK-cell-mediated lysis [161]. 

When activated, NK cells have the ability to generate in vitro and in vivo antitumor responses against GBM cell lines [111,162,163,164,165]. NK cells are capable of promoting regression of tumor spheroids [166], having the ability to infiltrate and induce apoptosis of LN-18, U87MG, T98G, and U251MG cell-line-derived GBM spheroids [167]. Given that tumor spheroids are 3D models that mimic several conditions from the tumor microenvironment (including tridimensional structure, nutrient and gas diffusion gradients, hypoxia, and low pH in the central region) [168], and considering specifically that GBM spheroids possess different patterns of gene expression compared to 2D cultures [167], the observed cytotoxic activity of NK cells suggests that, in vivo, NK cells might retain their killing effect even within the hostile GBM microenvironment. Furthermore, NK cells seem to preferentially kill GBM cancer stem cells (CSCs), at least in vitro [169,170,171,172,173]. If confirmed, this characteristic would be relevant, since CSCs have been deemed responsible for many of the more aggressive aspects of brain tumors, including heterogeneity, resistance to radiotherapy and chemotherapy, invasiveness, and angiogenesis [174]. 

Regarding the possibility of brain injury caused by adverse NK cytotoxicity against healthy cells, the literature indicates no major risks concerning direct neuronal or glial cytolysis. In fact, co-culture assays of astrocytes and activated human-derived NK cells showed a minimal killing effect compared to stem or differentiated GBM cells [173]. Data showing that NK cytotoxicity could be directed in vitro towards murine dorsal root ganglia neurons under several conditions, including hypoxic environments or infection [175,176,177], must be put in perspective by the observation that the killing of peripheral neurons by activated NK cells was inhibited by addiction of Schwann glial cells in triculture [178], suggesting that the cellular context surrounding peripheral neurons is important for determining NK killing activity. Furthermore, the cytotoxic activity was not reproducible in hippocampus or ventral spinal cord central nervous system neurons [175,177], indicating a tissue restriction. Moreover, despite the recently reported role of NK cells in the immuno-physiological regulation of adult brain neurogenesis and cognition, by killing of senescent neuroblasts [179], the number of neural stem cells seemed not to be affected by this activity [179], and in vitro assays indicate that GBM stem cells are preferentially targeted by activated NK cells in comparison to healthy human primary neural progenitors [133]. Given these preliminary observations, we will discuss the current progress and potential strategies for the therapeutic use of NK cells in GBM.

### 5.1. Adoptive Cell Therapy

Adoptive cell transfer (ACT) with NK cells is a strategy that involves the infusion of expanded and activated NK cells to have them act against the tumor. Currently, NK cells can be obtained at large scale through numerous methods. They can be isolated from the peripheral blood of patients, or from healthy donors (either from the postpartum placenta, the umbilical cord blood, or peripheral blood) [156]. Other approaches involve the differentiation of NK cells from stem cells—such as induced pluripotent stem cells (iPSCs), human embryonic stem cells (hESCs), or hematopoietic stem cells [156,180,181]—or the use of established human cell lines, such as the NK-92 cell line, averting the need for isolation and differentiation steps [182]. 

The activation and expansion of NK cells in vitro are important in order to achieve a significant representation of effector cells. This can be achieved by the use of cytokines such as IL-2 or IL-15, or through combinations of IL-12, IL-15, and IL-18; activating antibodies (such as OKT3) or “feeder” cells, such as K-562 cells, can also have this effect [156]. Furthermore, in the adoptive cell transfer, it is possible to genetically modify the NK cells by altering one or more of their characteristics [183], which is the foundation of the targeted therapy with NK cells carrying a chimeric antigen receptor (CAR-NK), which will be addressed below.

NK cells can be transferred in GBM intravenously and/or via the intracranial route [184]. The intracranial administration may involve, for example, the intracavitary application of cells right after the tumor resection surgery, the intracerebral or intrathecal injection (in the subarachnoid space), or the release of cells in the tumor cavity through surgically implanted intracranial devices (e.g., Ommaya catheters) [185,186,187]. In BALB/c-nude mouse models of GBM (using the U87MG line), the intratumoral injection of human NK cells required fewer cells than the intravenous administration (10^4^ vs. 10^7^ cells) to reach the same cell count in the tumor after 24 h. In comparison to the control group—treated with saline solution—injections promoted a 57–60% tumor volume reduction and an increase in the number of apoptotic tumor cells [139]. Despite the relatively simpler intravenous administration, the local application of NK cells seems more attractive since, at least in the model, it very significantly reduced the need for in vitro cell expansion. Though this advantage of the intratumoral ACT with NK cells over the intravenous route still needs to be confirmed, this study indicates that the injection of activated NK cells in the tumor might be an effective approach to treat GBM.

Some reports describing the use of NK cells derived from the patients (autologous) for the treatment of solid tumors have failed to demonstrate considerable therapeutic effects [188], and the development of new strategies has been delayed because of the practical difficulties involved in the production of highly purified and activated NK cells [189], as well as individualized and multiple-step protocols for each patient. However, new activation and expansion approaches have shown promising results in clinical reports. In a phase III cohort study published in 2019, patients with advanced colon carcinoma received autologous NK cells after the surgical treatment and oxaliplatin chemotherapy [190]. These NK cells were selectively expanded from the peripheral blood mononuclear cells (PBMCs), with 95% purity, via selective culture and stimulation [191]. The application of NK cells in multiple administration cycles did not lead to adverse effects, and significantly increased both the OS and the 5-year PFS in comparison with the control groups (72.5% vs. 51.7%, *p* = 0.037; and 51.1% vs. 35%, *p* = 0.044, respectively) [190]. In another recent phase I/II study by Nagai et al. (2020) [189], autologous NK cells were utilized to treat patients with various advanced solid tumors. Through the magnetic purification of the NK cells obtained from PBMCs, followed by culture in a growth medium supplemented with IL-2, it was possible to generate highly purified (>90%), expanded (>200-fold), and activated (>85% of CD69^+^ NKs, in 8/10 of the cases) NK cells. Intravenous administration cycles of 10^6^ to 10^8^ NK cells in the patients did not lead to severe adverse effects, and 4/10 patients showed signs of tumor control, with one patient presenting partial remission of a pulmonary metastasis and surviving for 4 years, and the other three patients presenting stable disease for an average period of 4 months. Collectively, the mentioned studies with solid tumors show evidence of safety, and of the potential of autologous NK cells to improve the clinical condition of the patients, which encourages studies in patients with GBM, where any effort to improve the prognosis is valid.

Indeed, in GBM, autologous NK cell therapy has been exploited in treatment with lymphokine-activated killer (LAK) cells [192,193,194]. LAK cells are a heterogeneous cell population—comprising mainly T lymphocytes, NK cells, and NKT cells—which are obtained through the activation and expansion of PBMCs with IL-2 [194,195]. Dillman et al. (2004) [185] conducted a phase I/II non-randomized study with patients with recurrent GBM who received autologous LAK cells intracranially after undergoing their second surgery. The patients with primary GBM who received LAK cell infusion had prolonged overall survival in comparison to the control group (mean OS: 17.5 vs. 13.6 months, *p* = 0.012). In this case, despite the seemingly positive effect of the immunotherapy and the in vitro cytotoxic activity against tumor cell lines, the NK cells represented only a reduced fraction of the cells expanded with IL-2 (CD3^−^ CD16^+^); thus, the relative contribution of the NK cells remained uncertain [185]. The same group subsequently conducted another phase I/II clinical trial with intracranial LAK cell therapy in a group of patients with recurrent GBM who mostly underwent conventional treatment, with surgery followed by chemoradiotherapy (TMZ) [194]. The patients who received the LAK therapy had a median survival of 20.5 months after the diagnosis [194], which is a longer time in comparison to the median survival of 15 months observed in patients treated with the conventional therapy [3]. However, only the number of T-LAK cells (CD3^+^ CD56^+^ CD16^+^) and the use of corticosteroids showed some correlation with the response, and no correlation was seen with the number of NK cells (CD3^−^ CD56^+^ CD16^+^) [194]. Only randomized large-scale studies would be able to determine the efficacy of therapy with LAK cells in GBM, and determine the relative contribution of NK cells.

Another alternative was tested for autologous NK cell therapy in patients with GBM and other high-grade recurrent gliomas who underwent tumor resection. In this case, the patients’ PBMCs were stimulated with the combination of irradiated feeder cells (HFWT) and IL-2 [184], leading to a selective expansion of the NK cells (median 85.9% purity). These cells had a more potent in vitro cytotoxic effect against GBM cell lines in comparison to LAK cells, which received only IL-2 [184]. When administered in patients, in combination with low intravenous doses of IFN-β, the therapy proved to be safe, and was associated with a reduction in tumor volume in some cases [184]. One patient presented stable disease until his death after 4 months, and another patient had a minor response, with 25% reduction of tumor volume within 4 months, and overall survival of 15 months after the treatment with the NK cells/surgery [184]. Even though the efficacy of the monotherapy with NK/LAK cells has not been proven, the results obtained thus far indicate that an improved strategy might achieve significant results.

Recently, the therapeutic potential of ACT therapy with NK cells activated with IL-2 and Hsp70 has been shown. The heat shock protein 70 (Hsp70) is a chaperone overexpressed in many cancers [196], including in human GBM [197,198]. In vitro treatment of NK cells with Hsp70 and IL-2 promotes cellular activation and greater antitumor cytotoxic activity [196], leading to the selective death of tumor Hsp70^+^ cells [199]. ACT with autologous NK cells activated with IL-2 and Hsp70 has been tested in patients with metastatic colorectal cancer or small-cell lung carcinoma (phase I), showing safety and potent cytotoxic effects in vitro [199]. In a murine model of GBM [164], the systemically injected IL-2/Hsp70-treated NK cells were able to efficiently cross the BBB and reach the tumor, establishing contact with the neoplastic cells. This was correlated with a reduction in the number of GBM cells (marked with GFP). Both the systemic and the intracerebral infusion of IL-2/Hsp70-treated NK cells led to the shrinking of the tumor to almost undetectable levels 8 days after treatment (*p* = 0.0029 and *p* = 0.0319, respectively), while non-activated NK cells were ineffective [164].

Allogeneic therapy, using NK cells derived from donors or from NK cell lines, despite expanding the sources of NK cells, has the disadvantage that allogeneic cells can be rejected by the patient’s immune system. To overcome this issue, lymphodepleting drugs—such as fludarabine and cyclophosphamide—can be administered days before the infusion of allogeneic cells [156]. The allogeneic strategy may be worthwhile, since in addition to averting the need for individual sample processing from each patient, it can eventually overcome the tumor suppression of autologous NK cells mediated through the expression of MHC class I molecules. This is possible because of mismatches between the iKIR receptors from the donor and the MHC class I molecules from the patient [200], leading to improved NK cell activation against the tumor. Several studies have shown that therapy with alloreactive NK cells is relatively safe, and does not lead to graft-versus-host disease (GvHD) [201]. This allogeneic rejection effect is already being exploited in the treatment of hematological malignancies [157], and there are strategies already available to potentiate alloreactivity, such as the screening of the donors [202] and the use of NK cells with only one KIR receptor, such as the NK-92 cell line [182], or sorted NK cells [203].

The FDA recently approved the use of allogeneic NK cells differentiated from hematopoietic stem cells from the human placenta (CNK-001) as a new investigational drug for the treatment of GBM. The approved clinical trial, NCT04489420, which is being conducted by the University of Texas MD Anderson Cancer Center, is the first attempt to evaluate monotherapy with allogeneic NK cells without genetic modifications in patients with GBM. The trial will evaluate the safety and preliminary efficacy of NK cells, comparing the intratumoral and intravenous routes, with the latter being associated with previous lymphodepletion with cyclophosphamide [204].

### 5.2. CAR-NK Cells

The success of chimeric antigen receptor (CAR) T-cell therapy against hematological tumors, with three therapies already approved for commercialization by the FDA, has prompted interest in improving the design of CARs, and in expanding this technique to other cell types, such as NK cells. CARs were initially developed to enable T cells to recognize and act against a given tumor antigen, bypassing the need for MHC presentation, which is necessary for recognition by the TCR [205]. CARs are engineered molecules composed of four main components (Figure 4): one single-chain variable fragment (scFv), derived from a monoclonal antibody, which binds to the antigen of interest, conferring specificity; the hinge or spacer, which gives support and flexibility to the scFv; a transmembrane domain; and one or more domains of intracellular signaling, responsible for cell activation [206]. In fact, this last component is used to classify the CARs in distinct generations, depending on the number of costimulatory domains they carry [206]. 

However effective they may be against neoplastic cells, CAR-T cells have some characteristics that hinder their efficacy. A significant drawback is that they still carry their own TCR, meaning that allogeneic T cells cannot be used to avoid GvHD. The use of NK cells to carry the CAR, on the other hand, would bypass this obstacle [159,207,208,209]. Initially, NK cell CARs displayed the same structure as those optimized and carried by T cells. However, although some costimulatory domains are common to NK and T cells—such as CD3ζ- and 4-1BB—other domains used in CAR-T cells, such as CD28, are not present in NK cells [210,211,212], meaning that the “net signaling” by these receptors had to be determined. The first-generation CAR-NKs only had signaling domains derived from CD3ζ [212,213,214]. Afterwards, in addition to CD3ζ, other costimulatory domains—such as CD28 [215,216], 4-1BB [217,218], 2B4 [218,219], and DNAM1 [219]—were added, giving rise to the second and third generations of CAR-NKs [212,220]. Curiously, even the CD28 costimulatory domain, developed for CAR-T cells, was also capable of triggering antitumor activities when used in CAR-NK cells. Illustrating the possibility and need for continuous improvement of CAR engineering, a study using anti-CD5 CARs showed that the NK-specific costimulatory domain 2B4 led to a superior performance when compared with the 4-1BB domain, even though 4-1BB is a domain present in both T and NK cells [218]. Therefore, the construction of CARs using other NK-specific domains can be a strategy to modulate the effects of CAR-NK therapy. Accordingly, adaptor molecules other than CD3ζ—including DAP10 [221] and DAP12 [222]—have already been tested, given that CD16, NKp30, and NKp46 receptors signalize via CD3ζ, NKG2D does so via DAP10 and KIR, and NKG2C and NKp44 do so via DAP12 [211,223]. Indeed, CARs bearing a DAP12 domain displayed improved antitumor activities in comparison to CD3ζ-based CARs [222,224], but CARs with CD3ζ were superior to those with DAP10 [221].

Thus, CAR-NK cells offer new possibilities of modulation for CAR-based strategies and, consequently, have the potential to overcome some of the hurdles found in CAR-T cell therapies. Indeed, while CAR-T therapies frequently cause high-grade adverse events—such as cytokine release syndrome and neurotoxicity [207,225,226]—these events are less frequent in CAR-NK therapies [159,207,208,209]. This is most likely due to the different cytokine release profile induced by these cells: CAR-T cells induce the release of pro-inflammatory cytokines, such as TNF-a, IL-1, and IL-6 [207]; in contrast, CAR-NKs, commonly induce the release of IFN-γ and GM-CSF [227]. As already mentioned, CAR-NK therapy also has the advantage of allowing the use of allogeneic cells without the occurrence of GvHD [157,228,229,230], overcoming this challenge described in CAR-T allogeneic therapies. On-target/off-tumor toxicity is also less frequent due to the limited persistence of CAR-NK cells in circulation [231]. Very significantly, the possibility of allogeneic CAR-NK therapy expands the range of NK sources that can be used, prompting a faster development of treatments. This is a great advantage in comparison to autologous therapy [211], in addition to being a more cost-effective strategy [208]. Hence, the safety profile and viability of CAR-NK therapy seem to be superior to those observed in CAR-T therapy, which would enable a reduction in the costs of this rather expensive treatment modality [211,212].

Furthermore, CAR-NK cells still preserve other activating mechanisms of NK cells [232], thus remaining able to target tumor cells via CAR-independent mechanisms. This further increases the efficiency of the elimination of neoplastic cells, because even if they no longer express the target tumor antigen—a common escape mechanism observed in CAR-T therapies [233,234]—or if the tumor has a heterogeneous antigen expression, CAR-NK cells will still be able to attack these tumor cells [235].

CAR-NK cell strategies are already being tested at clinical and preclinical scale for a variety of solid and hematologic malignancies, including CD19 for B-cell lymphomas [159], CD33 for acute myeloid leukemia [208], CD138 for multiple myeloma [236], HER2 for glioblastoma [237], mesothelin for epithelial ovarian cancer [238], and NKG2DL for colorectal cancer [224], EGFR for many solid tumors [239,240,241,242], ROBO1 for pancreatic carcinoma [243], glioma, and neuroblastoma [244], among others. Despite the growing interest in CAR-NK therapy, the greatest focus and most promising results are still observed for hematological cancers; solid tumors remain a difficult target, for reasons including their immunosuppressive tumor microenvironment. For glioblastoma, CAR-NK efforts are mainly focused on targeting EGFRvIII, EGFR, and HER2 [235].

The EGFR protein is minimally expressed or even absent in healthy cerebral tissue, but is overexpressed in 40–60% of glioblastoma tumors [245]. Approximately 20–40% of EGFR-expressing tumors also express the EGFRvIII variant form [246], which is the most common mutation occurring in GBM [247], and confers greater immunogenicity [215]. Murakami et al. (2018) [241] evaluated the engineering of the KHYG-1 NK cell line with an EGFRvIII-specific CAR, and observed the induction of a strong and significant inhibition of EGFRvIII-expressing U87MG GBM cell line growth, through the triggering of apoptosis. A further study by the same group was conducted in 2020 [248], showing the in vitro secretion of RANTES and—when recognition of U87MG EGFRVIII^+^ tumor cells by CAR-NK cells occurred—a significant secretion of TNF-α, IFN-γ, IL-2, and IL-6 cytokines. However, the treatment of immunodeficient mice bearing the specific target cell apparently failed to control tumor progression, and resulted in greater tumor occupancy, in comparison to PBS-treated mice. Since the immunohistochemistry analysis revealed an increase in necrotic cells at the tumor site, the authors suggest that observed growth in treated mice could be due to pseudoprogression, where an expansion of tumor volume due to an increased inflammatory infiltrate would mimic real tumor progression.

As CAR-NK immunotherapy strongly relies on the effective migration of these cells towards the tumor, Müller et al. (2015) [249] compared the function of EGFRvIII-specific CAR-NK cells with similar CAR-NK cells that were also engineered to contain the CXCR4 chemokine receptor. The incorporation of CXCR4 led to specific chemotaxis to CXCL12/SDF-1α-expressing GBM U87MG cells, in addition to significantly increasing the survival and even leading to complete remission of tumor xenografts in mice, in comparison to CAR-NK cells without CXCR4 co-expression.

Given the possibility of heterogeneous EGFRvIII expression in GBM, CAR-NK therapy with specificity to both EGFR and EGFRvIII has been addressed. Han et al. (2015) [240] showed that NK-92- or NKL-derived CAR-NK cells with specificity to both of these antigens displayed a robust increase in cytolytic capacity and IFN-γ secretion, in an EGFR-dependent manner, when co-cultured with GBM cell lines or patient-derived GBM stem cells (GSCs); furthermore, they efficiently suppressed tumor growth, and prolonged the survival of mice with orthotopic GBM xenografts. Genßler et al. (2015) [215] tested NK-92 cells expressing CAR with specificity to EGFR, EGFRvIII, or a shared epitope between these antigens, showing high NK cell cytotoxicity and specificity against primary GBM cells that expressed the antigens, as well as against GBM cell lines expressing EGFR and/or EGFRvIII. Interestingly, the administration of double-specific CAR-NK cells to NSG-immunodeficient mice transplanted with intracranial GBM xenografts led to prolonged survival, and prevented tumor escape by antigen loss more efficiently, when compared to monospecific CAR-NK cell therapy. 

The human epidermal growth factor receptor 2 (ErbB2/HER2) is a member of the EGFR family that is expressed in up to 80% of GBM cases, but not in normal neurons or gliocytes [250], and whose overexpression is correlated with earlier patient mortality [251,252]. Zhang et al. (2016) [237] evaluated the effect of NK-92/5.28.z cells—which are ErbB2-specific CAR-NK cells—both in NSG immunodeficient mice bearing orthotopic human GBM xenografts, and in C57BL/6 immunocompetent mice with murine subcutaneous or orthotopic GBM cells expressing ErbB2. The NK-92/5.28.z cells were able to lyse all of the ErbB2-positive cells analyzed in vitro and, after repeated intratumor injections, significantly prolonged the symptom-free survival of NSG mice bearing orthotopic xenografts of an ErbB2-expressing GBM in vivo. In C57BL/6 mice, the local therapy with NK-92/5.28.z cells eradicated the tumor in 4 out of 5 mice with subcutaneous tumors and in 5 out of 8 with orthotopic tumors. CAR-NK-cell-treated animals that eradicated the tumor rejected a rechallenge with GBM cells injected into the other cerebral hemisphere 120 days after the initial therapy, indicating long-term protection against tumor rechallenge at distant sites. Interestingly, rechallenged animals that rejected the tumor had significant levels of IgG antibodies against GBM cells [237].

A phase I clinical trial, CAR2BRAIN (NCT03383978), has been underway since 2017 at the Johann W. Goethe University Hospital, in Germany, to evaluate the safety profile, tolerability, maximum tolerated dose (MTD), maximum feasible dose (MFD), and antitumor effects of NK-92/5.28.z cell therapy against relapsed or refractory HER2-positive GBM in patients undergoing biopsy or tumor resection surgery. NK-92/5.28.z cells are repetitively injected in the resected tumor cavity, or centrally inside the tumor [253]. This is the first and only CAR-NK clinical trial registered on Clinicaltrials.gov analyzing CAR-NK therapy in patients with GBM.

The fact that only one clinical trial is investigating the potential of CAR-NK therapy in GBM, despite its clear potential, highlights the challenges still to be solved before its experimental results can be translated into clinical applications. One of these challenges is related to the efficacy of transduction methods used to insert the CAR receptor gene. The retroviral or lentiviral transduction in primary NK cells has poor effectiveness [254], but new approaches to overcome this difficulty are being developed, as is the case of the baboon envelope pseudotyped lentivirus (BaEV-LV) [255] and vectofusin-1 [256]. Currently, lentiviral transduction is the most used method in studies regarding CAR-NK for glioblastoma, since it appears safer than retroviral vector-mediated transduction [257]. Non-viral transduction alternatives are also being studied: electroporation leads to a product with short therapeutic window [211] and, thus, does not seem suited for clinical applications; the efficacy of transposons [258] and CRISPR/Cas9 [259] remains a work in progress. Some of the other challenges, already mentioned in the context of NK cell adoptive transfer, include the limited in vivo persistence after infusion, and limited proliferative potential.

For the treatment of GBM, both CAR-NK and CAR-T approaches still face additional challenges to overcome, given that this cancer is a solid tumor located in the central nervous system, which is one of the most critical, dynamic, and difficult-to-access regions of the body. One way to improve the in vivo persistence of CAR-NK cells in the suppressive tumor microenvironment could be by inducing the expression of stimulatory cytokines, such as IL-2, IL-7, IL-15, and IL-21. As we will better describe in the next section, these cytokines play a crucial role in promoting NK cell proliferation and survival [260]. Xu et al. (2019) [261] studied the co-expression of G2D antigen-specific CAR and IL-15 in NKT cells both in vitro, and in immunodeficient mice with neuroblastoma xenografts, observing that both NKT in vivo persistence and tumor control were improved. Liu et al. (2018) [216] also tested a similar strategy with umbilical-cord-blood-derived NK cells engineered to express a CAR specific to CD19, an IL-15 transgene, and an inducible caspase-9-based suicide gene (*iC9*), showing constitutive IL-15 expression by NK cells and functional improvement. The *iC9* gene’s pharmacologic activation promoted a fast NK cell depletion, which could be an interesting tool to use in the face of eventually less tolerable toxicities. Other strategies to enhance CAR-NK activity, currently under study, include the engineering of TGF-β receptors without their intracellular domains, in order to avoid the transduction of inhibitory signals [262] or, conversely, substituting these with costimulatory domains to promote NK cell activation [262,263]. These other approaches offer the prospect of eventually overcoming the immunosuppressive effect of the tumor microenvironment (TME) and obtaining more effective CAR-NK cells for the treatment of GBM. 

### 5.3. Cytokine Therapy

Cytokines influence all aspects of the immune response, and several have the potential to improve the function, proliferation, and survival of NK cells, and therefore can be used in NK-based immunotherapy, either by systemic administration or by in vitro incubation with NK cells in the process of generating the adoptive cells. IL-2, IL-12, and IL-15 are the main cytokines being explored in this context.

IL-2 was the first cytokine to be used in a clinical setting, aiming at what one could consider an improvement of the cytotoxic activity of NK cells against cancer, through the phenomenon of lymphokine-activated killer (LAK) cells [264]. Though IL-2 also affects the function of other cells—such as T and B lymphocytes—and, thus, the LAK phenomenon cannot be attributed solely to its action upon NK cells, they seem to be the major effectors among the LAK cells [265]. IL-2 induces the proliferation of NK cells, enhances their cytolytic capacity [266], and refines their signaling through activating receptors [156]. However, when administered in high doses, IL-2 induces severe adverse effects [264,267,268], in addition to promoting the expansion of CD25^+^ T_reg_ cells, which have immunoregulatory effects and compete with NK cells for binding with IL-2 [269,270]. To overcome the toxicity of IL-2, it can be administered in lower doses [158], and to improve its potential for immunotherapy, mutant forms of IL-2 with modified affinity for its various receptors have been generated, either with augmented affinity to the IL-2Rβ receptor, signaling effectively without the need for IL-2Rα (CD25), or even having different degrees of affinity to IL-2Rγ, thus having a range of signaling from full agonism to antagonism [271,272]. Moreover, IL-2 can also be used during the ex vivo expansion of NK cells [164], or after the adoptive NK cell transfer, in an effort to improve the survival and function of these cells in vivo [158,201,273,274,275]. For the GBM therapy, studies such as those conducted by Colombo et al. (2005) [276] and Qiao et al. (2018) [277] evaluated the combination of IL-2 with other therapeutic strategies, but mainly focusing on the effect promoted on T cells. Clinical studies involving the combination with other therapeutic strategies—such as expression of IL-13, resistance to glucocorticoids, and stem cell transplantation—have also been conducted for the treatment of brain tumors (NCT01082926, NCT00014573).

IL-15 is another cytokine closely related to the promotion of NK cell proliferation and survival [278,279,280], and has similarities with IL-2 regarding the binding to the receptor, the subsequent signalization, and biological activity [269,281]. However, IL-15 preferentially stimulates CD8^+^ T lymphocytes and non-terminally differentiated NK cells [282,283], without promoting the expansion of T_reg_ cells [284]. IL-15 signaling can occur through cis-presentation or trans-presentation, the latter being the most common. IL-15 trans-presentation via IL-15Rα promotes a higher immunostimulatory capacity in comparison to IL-15 alone [285], and the use of modified IL-15/IL-15Rα complexes [286,287] or the development of fusion proteins [288,289,290] leads to better results in comparison to rIL-15—for example, by enhancing its half-life [291]. The first clinical study using rIL-15 via intravenous infusions (bolus) for the treatment of cancers led to a considerable expansion of NK cells and TCD8^+^ cells, but with an intense cytokine secretion and dose-dependent toxicity [278]. Since then, several clinical studies have been conducted in order to better evaluate the administration of IL-15 alone [279,292,293,294] or in combination with other therapies [295] in different cancers.

In glioblastoma, IL-15 can have effects in different types of therapy [296,297,298], and regarding its effects in NK cells, Ma et al. (2021) [299] saw that the in vitro administration of a herpes-simplex-1-based oncolytic virus (OV) expressing the human IL-15/IL-15Rα sushi domain fusion protein (named OV-IL15C) was able to promote the secretion of soluble IL-15/IL-15Rα complexes by the infected GBM cells, inducing tumor cytotoxicity and greater survival of NK and TCD8^+^ cells. In animal models, this therapy led to increased survival and inhibition of tumor growth in the presence of TCD8^+^ cells. When in combination with EGFR CAR-NKs, the therapy increased tumor cell death in vitro, and in vivo it promoted the suppression of tumor growth and greater survival in comparison to monotherapy, leading to higher intracranial infiltration and activation of NK cells and TCD8^+^ lymphocytes, as well as elevated persistence of the CAR-NK cells. Mathios et al. (2015) [300] observed that the administration of the IL-15 superagonist complex ALT-803 in a murine model led to greater animal survival and durable antitumor response, but the simultaneous depletion of TCD4^+^ and TCD8^+^ cells abrogated the benefit in survival, whereas the depletion of NK cells alone did not affect the survival, suggesting that in spite of the presence of NK cells being detected infiltrating the tumor, their actions might not be crucial to the effect of this therapy in GBM. Interestingly, however, in other tumor models—such as bladder cancer—NK cells were responsible for the antitumor effect of the therapy with ALT-803 [301]. This is an observation that likely points to the very peculiar scenario of GBM, and which could help us to better understand its complex microenvironment. Another effect of IL-15 that may prove to be of translational value is that the antitumor potency of extracellular vesicles derived from NK cells also seems to be enhanced by priming with IL-15, leading to higher cytolytic activity and greater expression of molecules associated with NK cell cytotoxicity [302]. 

Another promising cytokine for cancer immunotherapy is IL-12, which presents several effects, being able to stimulate the production of cytokines by NK cells—such as IFN-γ, GM-CSF, and TNF-α [269]—to augment the cytolytic capacity of NK cells and TCD8^+^ lymphocytes, to promote the polarization of the TCD4^+^ lymphocytes to a Th1 profile [303], and to increase the ADCC against tumor cells opsonized with antibodies [304]. IL-12 has a narrow therapeutic window for its systemic administration [305]. Further studies have evaluated the effects of lower doses [306,307], the use of modified cytokines [308], and gene therapy or genetic engineering [269] as ways to reduce the adverse effects, in addition to evaluating the combination with other therapies [305], leading to some promising results. IL-12 therapy has already been tested for GBM in several studies [309,310,311,312,313,314,315,316], alone or in combination with other types of therapy. Among the studies that directly evaluated the effect of IL-12 therapy on the antitumor activity of NK cells in GBM, Chiu et al. (2009) [313] evaluated the effect of the administration of recombinant adeno-associated virus (rAAV) encoding IL-12 in nude mice with implantation of the DBTRG glioblastoma cell line. This treatment induced a substantial increase in the expression of IL-12, and an increase in the number of activated NK cells, with augmented cytotoxic activity and a decreased tumor growth rate compared to the control group.

Other cytokines, such as IL-18 and IL-21, also affect NK cells. IL-21, in addition to having effects on T and B lymphocytes [269], promotes NK cell survival and maturation, and induces proliferation, but can also lead to their apoptosis [317]. Preclinical studies demonstrated the promotion of antitumor activity by IL-21 [318,319] and an increase in NK-cell-mediated antibody-dependent cellular cytotoxicity (ADCC) [320]. Clinical studies demonstrated good overall tolerability, alone or in combination with other agents, with modest but still promising results [321,322,323,324,325,326]. In glioblastoma, IL-21 gene transfer in mice resulted in a greater antitumor response from NK cells [327]. The use of IL-21 in the ex vivo stimulation of CD8^+^ and γδ T lymphocytes before adoptive cell transfer therapy has also been assessed [328,329]. IL-18 can assist in the survival of NK cells and enhance their functions, acting in synergy with IL-12 to increase IFN-γ secretion [330,331,332]. In fact, pre-activation of the NK cells used for ACT therapies with IL-12, IL-15, and IL-18 is capable of improving the persistence, function, and antitumor activity of these cells in mice, when compared to the pre-activation with IL-2 or IL-15 alone [333].

### 5.4. Monoclonal Antibodies, Killer Cell Engagers, Immunocytokines, and Other Antibody-Based Strategies to Boost NK Activation

A broad range of antibody-based therapies have been tested and approved for the treatment of various cancers. Since some of them can trigger NK cell activation, they can be used to overcome the dysfunction and immunosuppressed status of the patients’ NK cells, or even to improve adoptive cell therapy efficiency. One strategy consists of the use of monoclonal antibodies (mAbs) targeting tumor cells, rendering them susceptible to ADCC—a process that can be mediated by NK cells [334]. The cetuximab antibody, which blocks EGFR and EGFRvIII signaling [335], already has a confirmed effector mechanism through ADCC against GBM cells [336], and its activity in patients with GBM is under investigation [337]. To enhance ADCC, another target for modulation is the Fc portion of antigen-specific monoclonal antibodies, which can be engineered [338,339,340] or glycosylated [341,342,343]. Schmied et al. (2019) [344] evaluated the effect of S239D/I332E amino acid exchange in an mAb reactive to CD133 (293C3-SDIE) in a preclinical model of colorectal cancer. This modified mAb had an increased CD16 affinity and induced a robust NK cell activation, triggering their degranulation, IFN-γ production, ADCC activity, and killing of the tumor cells [344]. Clinical trials have addressed Fc modifications of mAbs for many cancers, including acute myeloid leukemia (the recently concluded NCT02789254), B-cell acute lymphoblastic leukemia, and breast cancer, where they demonstrated good tolerability but reduced effectiveness as a monotherapy [345] (NCT01828021). This scenario suggests that, at least for these mentioned approaches, multitarget therapy could be a more effective option.

Combined regimens of monoclonal antibody therapy with costimulatory cytokines—such as IL-2, IL-12, IL-15, IL-18, and IL-21—are another approach to improve ADCC activation of NK cells [44,269]. This combination can be also made through the use of immunocytokines, which are composed of a monoclonal antibody linked to a cytokine, aiming to assist in the activation of immune cells. By doing this, the antibody can bind to a tumor antigen, promoting the directed delivery of the cytokine into the tumor microenvironment, thus preventing the adverse events associated with systemic administration of cytokines while also activating tumor-infiltrating leukocytes, including NK cells [346]. For GBM, Weiss et al. (2020) [347] tested the immunocytokine strategy by linking an L19 antibody, which is specific to a tumor-associated epitope of extracellular fibronectin (expressed in the tumor stroma of most brain cancers, but generally not in healthy tissues), to IL-2, IL-12, or TNF-α, and administering the immunocytokine in immunocompetent mice bearing GL-261 or SMA-560 glioma cell lines. The mAb L19 achieved high concentrations in the tumor microenvironment and, when linked to any of the cytokines and injected intravenously, induced tumor shrinking in the animals. L19-mTNF or L19-mIL-12 treatments also promoted the release of other pro-inflammatory cytokines and accumulation of lymphocytes in the tumoral niche, in addition to significantly improving survival. Conversely, L19-mIL-2 therapy had modest effects on the outcome, and was deemed to depend on adaptive immunity since, when applied to RAG-deficient mice—who lack T, B, and NKT cells—it lost its therapeutic effects. L19-mIL-12 therapy increased TCD4^+^ and TCD8^+^ but also NK infiltration, in addition to activation of the latter in mice bearing both types of glioma cell tumor; L19-mTNF treatment mainly improved NK cell activity. The administration of L19-mTNF or L19-mIL12 immunocytokines in combination with radiotherapy and chemotherapy demonstrated synergistic effects in murine GL-261 glioma models. Given these promising preclinical results, the same group initiated a phase I/II clinical trial in 2018 to evaluate the effects of the systemic administration of a fully human L19-TNF immunocytokine in patients with grade III or IV and IDH1 wild-type gliomas undergoing their first relapse (NCT03779230). Until now, this therapy has been safe, and correlated with increased tumor ischemia and necrosis. A biopsy sample derived from a patient with progressive disease showed high levels of CD4 and CD8 T cells in the tumor after the treatment, in addition to increased staining with apoptosis markers.

Another possible strategy aiming at the improvement of NK cell function is the development of bi-specific or tri-specific killer cell engagers (BiKEs or TriKEs), which are composed of a single-chain variable fragment (scFv) of an antibody connected to another one (BiKE) or two (TriKE) scFv portions derived from antibodies with different binding specificities [348,349]. In this way, several fragments can be arranged to compose multifunctional molecules. For example, in order to promote NK-cell-mediated tumor cell death through the formation of immunological synapses and ADCC [350], it is possible to make BiKEs with specificity to a tumor antigen of choice and to the CD16 expressed in the NK cells. Other activating receptors can be targeted, such as NKG2D and NKp46 [348]. The TriKEs can use the same strategy, but with the benefit of an additional scFv domain, allowing the engagement of more than one receptor. Interestingly, CD16- and NKp46-reactive TriKEs were more capable of mediating NK cytotoxicity against tumoral cells in comparison to the mixture of specific BiKEs to each receptor [348]. One variation of this approach is the replacement of the scFvs with a cytokine, such as IL-15 [351], with the purpose of promoting greater expansion, viability, function or in vivo persistence of the NK cells [16,352]. BiKEs and TriKEs have other advantages, such as providing a stronger interaction with anti-CD16 than the one observed in natural CD16-Fc antibody interactions [353], having reduced size (50–75 kDa) in comparison to bi- and tri-specific antibodies (300–450 kDa) [354], which leads to a better biodistribution needed for the treatment of solid tumors, in addition to being non-immunogenic [350]. BiKEs and TriKEs have already been developed and evaluated for a broad variety of antigens from different tumors, such as CD30 for Hodgkin’s lymphoma [355,356], HER2 for breast cancer [357,358,359], CD19 for non-Hodgkin’s lymphoma cells [360,361,362], CD33 or CD33 together with CD123 for acute myeloid leukemia [363,364] (NCT03214666), EpCAM for carcinomas [365], CD133 for colorectal cancer [366], EGFR for carcinomas [295,367,368], and B7-H3 for a variety of solid tumors [369]—but not yet for GBM. Further alternatives that have been already discussed in the literature include the blockage of immune checkpoint receptors via scFvs, and scFv blocking of TGF-β or ADAM-17— the later, a disintegrin and metalloproteinase involved in CD16 shedding [349].

Yet another strategy with a similar rationale is the use of recombinant immunoligands, which are fusion proteins containing a variable portion (scFv) with specificity to a tumor antigen fused to the C-terminal portion of the NK-activating ligand. Strategies under study include, for example, immunoligands containing the portion of the ligand ULBP2, which activates the NKG2D receptor, or the B7H6 molecule, which activates the NKp30 receptor. In this way, the recognition of the tumor cell and the activation of the NK cell are facilitated [370,371].

Blocking inhibitory receptors displayed on T and NK cells using monoclonal antibodies (checkpoint blockages or checkpoint inhibitors) might also bring good results for GBM. Indeed, anti-CTLA-4 (i.e., ipilimumab), anti-PD1 (i.e., nivolumab), and combined therapies with these two antibodies [372,373] have brought very good results in different cancers, and one could say that they have changed the clinical perspective on immunotherapy as a whole. For patients with recurrent GBM and undergoing standard therapy, who were enrolled in phase I of the Checkmate 143 clinical trial (NCT02017717), treatment with ipilimumab showed milder adverse effects than its combination with nivolumab, although both treatments were found to be tolerable [374]. Subsequently, the same study provided the results of the phase III trial, including 369 randomized individuals. The safety profile was consistent with known symptoms of the therapies, mainly comprising hypertension and fatigue; the efficacy analyses showed similar overall survival of both regimens (9.8 and 10 months for nivolumab and bevacizumab, respectively), in addition to prolonged progression-free survival in the nivolumab group (3.5 vs. 1.5 months, *p* < 0.001). An exploratory analysis suggests that the basal corticosteroid level and the methylation status of the MGMT promoter could have prognostic value, but this claim awaits validation of he results [375]. Recently, Shevtsov et al. [376] showed evidence indicating that the use of checkpoint inhibitors can improve the efficacy of NK-cell-based immunotherapy for GBM. In this case, NK cells activated with IL-2 and the TDK peptide (derived from Hsp-70) displayed a more potent cytolytic effect against GBM cells (GL261) and other tumors in vitro when associated with an anti-PD1 antibody. The best in vivo outcome in immunocompetent mice carrying GL261 GBM cell line tumors was observed in anti-PD1 + NK combined therapy, considering tumor shrinkage and overall survival in comparison to single regimens [376]. As for mice with lung cancer (A549) and treated with human activated NK cells and humanized anti-PD1, there was a prominent increase in survival and the highest degree of NK cell and TCD8+ lymphocyte infiltration in the tumor when compared to the other groups [376], strongly suggesting that PDL1/PD1 interaction can be advantageous in combination with NK cell adoptive therapies. However, the efficacy of the PD1/PD-L1 axis in NK immunotherapy was also contested by preclinical data, given no additional effects of PD1 blockade in terms of in vitro killing of GBM lines or overall survival in a murine immunodeficient model bearing U87MG-derived subcutaneous xenotransplants [111].

In addition to the previous strategy, the blockage of other inhibitory axes has been tested. Antibodies that block the NKG2A receptor (e.g., monalizumab) [377] or inhibitory KIR receptors, such as KIR2DLs (1-7F9) [378,379], have already been assessed in humans and other animals. These blockages are advantageous because they exceed the suppression caused by the tumor on autologous NK cells and T lymphocytes via class I MHC molecules, which is known to occur in GBM. The combination of monalizumab and cetuximab (anti-EGFR) potentiates the NK cells and LTCD8 ADCC in vitro, and showed safety and potency for disease control in preclinical models and in patients with head and neck squamous-cell carcinoma (SCCHN) [377]. Here, once more, the combination of different immunotherapeutic strategies seems to promote synergistic actions, recapitulating what one would expect from a “physiological” immune response, which is always dependent on the integration of various effector mechanisms. Additionally, and in the same direction, other NK-cell-inhibitory receptors—such as T-cell immunoglobulin and mucin domain 3 (TIM-3), and lymphocyte activation gene-3 (LAG3)—are now being investigated, alone or in combination with other therapeutic approaches, for the immunotherapy of a wide spectrum of cancers [51], including GBM (NCT03493932, NCT02658981, NCT03961971). 

One alternative strategy to the blockage of NK cell checkpoints is the use of the CRISPR/Cas9 genetic editing system. Analogous to the aforementioned antibody-based approaches, CRISPR/Cas9 could be a powerful tool in adoptive therapy for impairing NK cell inhibition and, consequently, boosting their activation and antitumor action. The disruption of checkpoint molecules by genetic editing in T cells using CRISPR/Cas9, addressing targets such PD-1 and CTLA-4 pre-clinically, showed the potential of its clinical exploitation [380,381,382]. In the same direction, clinical trials have reinforced the feasibility and safety of T-cell adoptive therapy in the treatment of patients with non-small-cell lung cancer [383], refractory cancers [384], and esophageal cancer (NCT 3081715, completed, not published). Other trials with the same objective are in progress (NCT0374796, NCT 03545815). In this scenario, some targets have been recently searched in NK cells, alone or in combinatorial settings (multiplex), as a proof of concept for the molecular function, methodological efficiency, and feasibility of NK immunotherapy designs, including TIGIT, CD96, DNAM-1, NKG2A, and TIM-3 [259,385,386], along with other regulators of NK cell activity [173,387]. 

Indeed, the selective CRISPR/Cas9-driven disruption of PD-1 in human primary NK cells enhanced the production, degranulation, and target cell apoptosis of IFN-γ against different tumoral cell lines with variable PD-L1 expression [388]. Regarding specific studies of GBM, Morimoto et al. (2021) [386] separately tested the knockout of two exons of the *TIM3* gene in primary human NK cells, via the electroporation of the Cas9-guided RNA ribonucleoprotein complex. This methodology resulted in selective target mutation and efficient in vitro inhibition of the growth of T98G and/or LN-18 GBM cell lines [386]. On the other hand, CRISPR/Cas9-edited CD96^−/−^ human NK cells did not display significant alterations in IFN-γ and TNF-α production, degranulation, or cytotoxic activity following stimulation with the U261MG CD155^+^ GBM line, in contrast to other tumor cell lines used. These observations, however, do not necessarily imply that CD96 NK knockout is ineffective, since primary lines and in vivo studies would be required for proper validation [385]. 

In addition to the most common immune checkpoints, other regulators have been linked to hypofunction of tumor-suppressed NK cells, indicating possible candidates for targeting in immunotherapy settings. This is the case of CD9 or CD103 TGF-β-inducible tetraspanins (αv-integrin ligands), for example. In a recent study, Shaim et al. (2021) [173] elucidated the role of αv–integrin interaction and subsequent TGF-β production by human GBM stem cells as evasion strategies employed to induce NK cell dysfunction. The knockout of CD9 or CD103 in human NK cells prevented immune dysfunction after GBM stem cell co-culture, partially restoring the cytotoxic capacity of NK cells against K-562 target cells in vitro; the cytotoxicity was completely restored in the case of the double-knockout [173]. Other important preclinical studies have focused on the CIS intracellular protein, which negatively regulates IL-15 signaling in murine and human NK cells [389,390]. CRISPR/Cas9-directed deletion of the cytokine-inducible SH2 gene (*CISH)* in murine NK cells resulted in hyperresponsiveness of *CISH*^−/−^ NK cells subjected to IL-15 stimulation, with increased expansion rates in comparison with wild-type NK cells, enhanced cytotoxicity towards tumor cells in vitro, and reduction in metastasis in melanoma-, prostate-, and breast-cancer-transplanted mice [389]. Additionally, based on preliminary preprint data, the importance of CIS seems to be reproducible in human NK cells using CRISPR/Cas9-based loss-of-function in vitro experiments [390]. 

Before their translation, however, CRISPR/Cas9 approaches must overcome technological hurdles, such as the means of delivering Cas9 into the cell. Lentiviral transfection has poor transfection efficiency, even at high viral titers [391] and both primary and NK-92 line NK cells are relatively resistant to plasmid-based editing, yielding poor efficiency and cytotoxicity [382,392]. However, nucleofection of preassembled Cas9-guided RNA complexes was reported as a relatively efficient and minimally toxic method [259,385,392]. Nevertheless, methodological optimization may lead to application of CRISPR/Cas9-based checkpoint knockout in NK cell clinical trials, thus opening other possibilities of treatment for cancer patients. 

### 5.5. Tumor Sensitization

The efficacy of adoptive therapies might be increased by the previous sensitization of the GBM cells to the cytotoxicity mediated by the NK cells. The proteasome inhibitor bortezomib (BTZ), which is an antineoplastic agent already approved and commercialized for the treatment of multiple myeloma and mantle-cell lymphoma [393], is an eligible candidate for this function; data regarding its relation to the anti-GBM activity of NK cells have been reported recently by Navarro et al. (2019) [161] and Luna et al. (2019) [394]; both studies showed that primary GBM cell lines, when treated with BTZ, had a significantly increased membrane expression of ligands for the NK-activating receptor NKG2D, and also for the death receptor TRAIL-R2. Furthermore, Navarro et al. [161] showed that BTZ could make GBM cells more susceptible to NK-mediated lysis via TRAIL, since at least some of the GBM cell lines tested showed greater death rates when treated with the combination of BTZ and NK cells than when treated with NK cells alone. However, in humanized mice with patient-derived GBM tumors, no statistically significant differences were noted between the intralesional monotherapy with autologous NK cells and the combined therapy with BTZ, even though the only two animals that remained alive by the 124th day were the ones that received the combined NK + BTZ therapy. The authors explained this failure as being the result of the toxic effect of BTZ on NK cells observed in vitro [161]. On the other hand, Luna et al. (2019) [394] used intralesional therapy with heterologous NK cells after the systemic administration of BTZ in mice with the U87MG cell line tumors. This therapy led to a significant decrease in tumor volume in comparison to the NK therapy or BTZ alone, also showing the longest effect in the control of tumor growth. Another potential effect of BTZ demonstrated in vitro is the induction of GBM stem cells’ susceptibility to NK-mediated lysis, which would be a very interesting effect for tumor control [394].

Another potential agent is the histone deacetylase inhibitor (HDACi) trichostatin A (TSA) [395]. In addition to its antineoplastic direct effects against a fraction of the GBM cells, through the promotion of acetylation and cell death, TSA was also able to induce the expression of ligands for the NKG2D- MIC-A and ULBP2- receptors, at both protein and mRNA levels, making these cells susceptible to NK-mediated lysis via TRAIL. The blockage of NKG2D with monoclonal antibodies significantly reduced the lysis effect, evidencing its role in this setting. Further confirming the role of NK cells in the antineoplastic effects of TSA, depletion of NK cells reduced GBM susceptibility to TSA treatment, both in vitro and in vivo [395].

Significantly, temozolomide (TMZ), which is used in the standard therapy for GBM, has also been proven to induce the expression of NKG2D (ULBP2) ligands on the surface of GBM cells, sensitizing them to Tγδ lymphocyte killing via both NKG2D and TCR [396]. Specifically concerning the synergistic effect of TMZ and activated NK cells, inhibition of GBM growth in vitro, in a cell-line-dependent manner, was previously reported, but without additional effects on tumor cell apoptosis and senescence [165]. Nevertheless, the administration of TMZ was concomitant—not before NK exposure—limiting the evaluation of the occurrence of prior tumor sensitization. Therefore, the sensitization activity of TMZ to NK cell killing should be investigated in further studies.

Given the cytotoxic function of these chemotherapeutic agents on NK cells, their use might prove to be more effective in adjuvant settings, with administration after the standard GBM treatment, but prior to adoptive transfer of NK cells. Nevertheless, additional studies with these or other chemotherapeutic agents may unveil treatment regimens capable of both directly killing GBM cells and potentiating their sensitivity to immune effectors, such as NK cells. 

## 6. Conclusions

Despite the constant efforts over the past decades, GBM remains a challenging disease, and the prognosis of patients is still poor. Key characteristics of GBM—such as its intrinsic heterogeneity and the strategies it displays to overcome the immune system, which include several mechanisms of NK cell immunosuppression—make it partially resistant to the various antineoplastic treatments employed.

NK cells are lymphocytes of the innate immune system with particularities that could make them advantageous in antitumor responses: their natural cytotoxicity against neoplastic cells, their numerous helper functions and, consequently, their important contribution to immunosurveillance, and to response against established tumors. In agreement with this, NK cell immunotherapy has proven to be a powerful tool for the treatment of various experimental tumors, and initial clinical data from studies in GBM indicate a good safety profile and effectiveness for NK-cell-based immunotherapy. 

The arsenal of strategies involving NK cells against GBM cells is certainly not limited to those mentioned here, since an increasing number of studies have been showing promising results, and the therapeutic possibilities are being radically expanded. Future strategies will benefit from screening for biomarkers associated with therapeutic efficacy, the combination of multiple therapeutic approaches and, ideally, personalized treatment regimens, considering the unique features of each patient. Thus, even if the contribution of NK cells for antitumor immune response in GBM patients still requires further study, they definitely appear to be a very promising and interesting component of the immune system’s response against GBM, which could be harnessed to improve treatment and, thus, patient care.

## Figures and Tables

**Figure 1 biomedicines-10-00400-f001:**
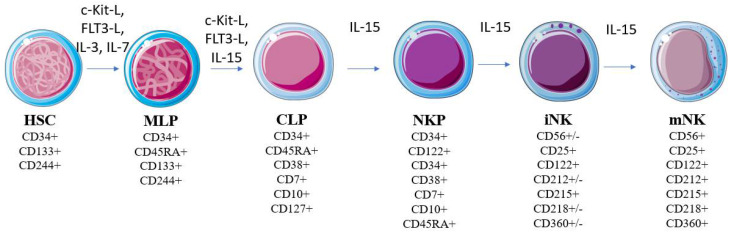
Model of human NK cell development. NK development from HSCs is regulated by multiple cytokines (e.g., FLT3-L, c-Kit ligand, IL-3, IL-7, and IL-15). Modifications of the molecules’ expression patterns are correlated with developmental stages. Abbreviations—Flt3l: FMS-like tyrosine kinase 3 ligand; c-Kit: c-Kit ligand; IL: interleukin; HSC: hematopoietic stem cell; MLP: compromised multipotent lymphoid progenitor; CLP: common lymphoid progenitor; NKP: NK cell lineage precursor; iNK: immature NK cell; mNK: mature NK cell.

**Figure 2 biomedicines-10-00400-f002:**
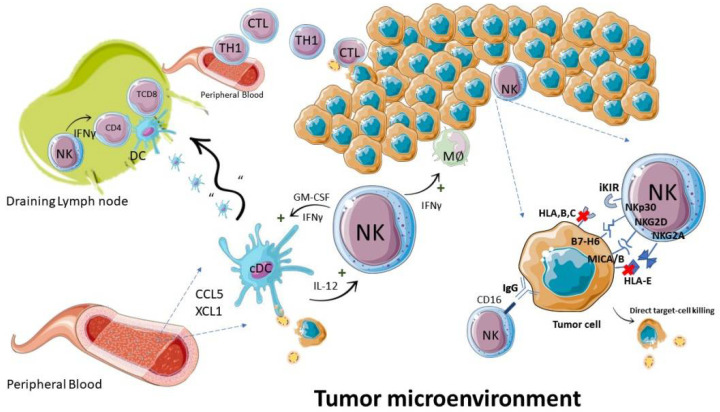
NK cells during an antitumor immune response. The antitumor function of NK cells is executed by direct killing of neoplastic cells and by regulation of immune cells: such as induction of MØ: TH1: and dendritic cell pro-inflammatory responses. Abbreviations—IFN: interferon; CTL: cytotoxic T lymphocyte; TH1: T helper type 1 lymphocyte; cDC: conventional dendritic cell; MØ: macrophage.

**Figure 3 biomedicines-10-00400-f003:**
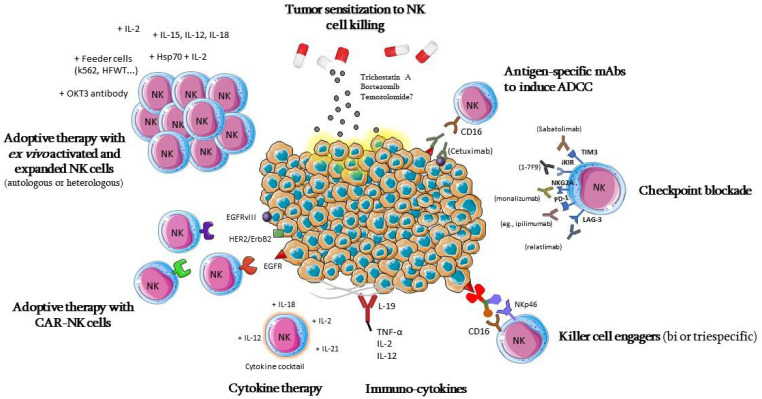
Strategies of NK-cell-based immunotherapy for the treatment of GBM.

**Figure 4 biomedicines-10-00400-f004:**
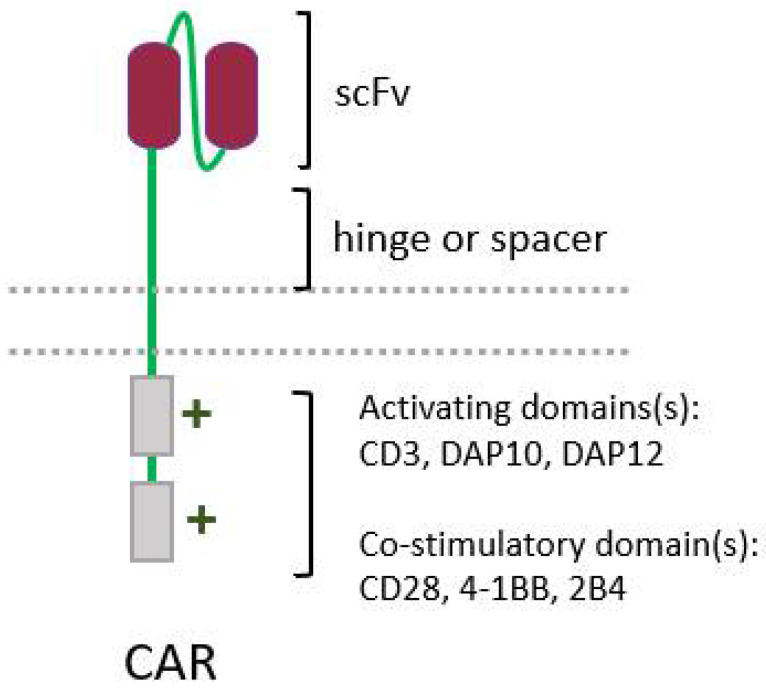
Simplified structure of a conventional chimeric antigen receptor (CAR), showing its four main elements: a single-chain variable fragment (scFv), the hinge, and the intracellular activating domain(s).

**Table 1 biomedicines-10-00400-t001:** Main human NK-cell-activating and -inhibitory receptors and their ligands.

Receptor	Ligand(s)	Function	Reference
NKp30	B7-H6, BAT3, viral ligands, heparan sulfate proteoglycan	Stimulatory	[42,43]
NKp44	Viral ligands, heparan sulfate proteoglycan, nidogen-1, PCNA	Stimulatory	[42,43]
NKp46	Viral ligands, heparan sulfate proteoglycan, vimentin	Stimulatory	[42,43]
FcγRIIIa	Fc portion of IgG	Stimulatory	[44]
DNAM-1	PVR (CD155), nectin-2 (CD112)	Co-stimulatory	[45]
CD94/NKG2A	HLA-E	Inhibitory	[46]
CD94/NKG2C, E	HLA-E	Stimulatory	[47]
NKG2D	ULBP1-6, MIC-A, B	Stimulatory	[48]
KIR2DS(1-5)/KIR3DS1	MHC class I	Stimulatory	[49]
KIR2DL(1-5)/KIR3DL(1-3)	MHC class I	Inhibitory	[49]
2B4	CD48	Co-stimulatory/inhibitory	[50]
PD-1	PD-L1, PD-L2	Inhibitory	[51]
CTLA-4	CD28	Inhibitory	[51]
TIGIT	PVR (CD155), nectin-2 (CD112), nectin-3 (CD113), nectin-4	Inhibitory	[52,53]
LAG-3	HLA class II, galectin-3	Inhibitory	[51,54]
TIM-3	Phosphatidylserine, HMGB1, CEACAM1 glycoprotein, galectin-9	Co-inhibitory	[51,55]
Tactile (CD96)	PVR (CD155), nectin-1 (CD111)	Co-inhibitory/stimulatory	[56]

**Table 2 biomedicines-10-00400-t002:** Some reports of tumor-infiltrating NK cells (TINK) in the GBM microenvironment.

Method	Markers	Infiltration in GBM	Main Phenotypic and Functional Alterations	Sample Size	Reference
Flow cytometry	CD14^−^ CD3^−^ CD56^+^	2.11 ± 0.54% of leucocytes	Predominance of CD56^dim^ CD16^−^ NK cells. Some NK cells lack expression of NKG2D (42.55%)	*n* = 8	[128]
Flow cytometry	CD19^−^ CD3^−^ CD56^+^ CD16^high^, among others	1 ± 5% of lymphocytes and 0.05 ± 0.05% of all cells in the tumor sample	Not measured	*n* = 53	[129]
Flow cytometry	CD45^+^ CD3^−^ NKp46^+^	Not measured	↓ Activating receptors’ protein levels compared to PB NK cells (NKp30, NKG2D, DNAM1, and CD2); ↑ CD69, a marker of activation; ↑ CD9, a TGF-β-induced molecule, described previously as being related to impaired activation of NK by modulation of NKG2D levels	*n* = 8 for NKp30, CD69, NKG2D and CD2) *n* = 9 for DNAM-1 *n* = 5 for CD9	[133]
Mass cytometry (CyTOF)	CD3^−^ CD56^+^ CD16^+^	~10% of leucocytes, with no significant difference from PB NK cells.	↑ CXCR3 protein and ↓ IFN-γ transcripts compared to PB NK cells	*n* = 8	[134]
RNA-Seq - CIBERSORT (In silico analysis)	13 genes’ expression profiles	9.8% of deconvoluted leukocytes	More quiescent than activated NK cells	*n* = 540	[132]
Immunohistochemistry	CD56^+^	~50% of samples showed a rare level of GBM intratumoral CD56^+^ cells (<5 NK cells in 10 high-power fields (HPFs)) The lasting % of samples showed focal (2–20 cells/HPF) or median (20–100 cells/HPF) infiltration, but not extensive infiltration (>100 cell/HPF) in intratumoral tissue; 49% of samples showed some level of perivascular NK infiltration, and 89% of samples showed some intratumoral infiltration	Not measured	*n* = 63	[127]

Abbreviations: ↑—an increasing in the molecule(s)’s expression; ↓—a decreasing in the molecule(s)’s expression.

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
