# Peer review of "The War Is on: The Immune System against Glioblastoma—How Can NK Cells Drive This Battle?"

_biomedicines, 2022, doi:10.3390/biomedicines10020400_

Round 1

Reviewer 1 Report

Dear chief editor,

This study is a detailed evaluation of the effects of NK cells on glioblastoma based on a vast number of articles. The review is clearly described and this content will be educational and informative for readers in neuro-oncology.

MAJOR POINTS

  1. The strategies to knockout the checkpoint molecules of primary NK cells using genome editing technologies such as CRISPR/Cas9 are beginning to be reported. Should be  summarized this point. See below the references. These are not all.

https://doi.org/10.1084/jem.20201529

https://doi.org/10.1038/s41416-019-0708-y

https://doi.org/10.3324/haematol.2020.271908

https://doi.org/10.1002/JLB.2MA0620-074R

http://dx.doi.org/10.1136/jitc-2020-001975

https://doi.org/10.1172/JCI142116

https://doi.org/10.3390/ijms22073489

  1. L351

 Authors mentioned low intratumoral infiltration of NK cells in GBM microenvironment. The authors should summarize how NK cells administered intratumorally or directly reach the tumor and how they show antitumor effects in the tumor.

  1. L890

Authors discussed checkpoint inhibitors for GBM in this paragraph. But, they didn’t concern the past clinical trials. I would recommend you to add the important paper (The CheckMate 143 trial NCT02017717)

Some study reported negative data with NK cells combined with anti-PD1 antibody for PD1/PD-L1 pathway in GBM.

https://doi.org/10.1001/jamaoncol.2020.1024

https://doi.org/10.3390/ijms22189975

https://doi.org/10.1172/JCI133353

  1. “(Error! Reference source not found.)” is often found. Please insert the appropriate reference.

MINOR POINTS

Figure1: I think that figure 1 is duplicated on page3. Please clarify figure1.

Line 57:

"null cells" was described in Ref 9-11, but these references did not define NK cell were "null cells". This term was defined in the below reference.

https://doi.org/10.1002/eji.1830090204

Line 84-85:

Ly 49, NK1.1 description were not included in Ref 20, and Ref 19 was not directly cited about these molecules. It would be better to change into the following reference.

https://doi.org/10.1016/S0960-9822(95)00194-1

In addition, CD49 description could not be found in Ref 19 and 20. Are these references correct?

Line 85-86:

Definition of human NK cells were described in Ref 16, but not directly citated.

It would be better to change the following reference.

https://doi.org/10.1084/jem.169.6.2233

https://doi.org/10.1016/S0065-2776(08)60845-7

Line 102   It could not be found any mention of the NKP80 in Reference 36. Is it correct?

Line 108  Is Fl3tl correct?

Table 1  Recommend following suggestion:

CD96 should be added in this table.

Change CD94::NKG2A into CD94/NKG2A

Change CD94::NKG2C/E into CD94/NKG2C,E

Other ligand of TIGIT and TIM3 were reported. Please add.

Line 120-128  Please add DNAM-1description.

Line 129-130

It would be better to add CD96 description.

Line 137

It would be better to changed CD94:NKG2A/C into CD94/NKG2A, C.

Line 169

Contact[62] Insert space between “contact” and “[62]”.

Line 172-176 Should be added RANTES description.

Line 252  Should be added HIF1-alpha description.

Line 264 Is FLTL3 correct?

Line 292: Author’s name is not true. “Nersesian” is true.

Line 324 What is TINKs?

L365: "iKIR" appears for the first time in this text. “iKIR” means inhibitory KIR? Please make it clear.

Line 399

The following references also reported about antitumor effect of NK cell against GBM cell lines. Please add.

https://doi.org/10.1371/journal.pone.0212455

https://doi.org/10.3390/ijms22189975

Line 400

The following reference is described about NK cell mediated cytotoxicity against GBM spheroids. Please add.

https://doi.org/10.3390/cancers13194896

Line 621   Is KHTG-1 correct? KHYG-1?

Line 620-623

Murakami et al. further reported in vivo study. Please add following reference.

https://doi.org/10.21873/anticanres.14304

Line 702: "TME" appears for the first time in this text. What dose “TME” means ? Please make it clear.

Line 792: "ADCC" appears for the first time in this text. “ADCC” means “antibody dependent cellular cytotoxicity”? Please make it clear.

Line 802 Line break is required.

Reviewer 2 Report

The authors present an informative review of NK cells activity in GBM

a few comments

1)figure 1 is duplicated in the manuscript

2)on page 22 the authors indicate nk cells are proved to modulate TMZ activity and then in in the following sentence indicate there requires more clinical evidence.  The reviewer suggests removing such language as it appears speculative how NK cells would alter responsiveness to a DNA damaging agent

3) could the authors mention any co culture study that involves mixing NK cell with developing astrocytes and neurons? This would more specifically address the “BBB” as well as support or refute claims made speculating that NK cells would spare normal brain function in a clinical/therapy based environment 

Round 2

Reviewer 2 Report

Thank you for your revisions

please remove the word proved in. ….also been “proved” to induce. 1080

Please reference TMZ as opposed to “the drug” line 1082

please indicate the term “cell dependent manner” on line 1083 as opposesd to line dependent